# Short-term association between ambient air pollution and cardio-respiratory mortality in Rio de Janeiro, Brazil

**Taísa Rodrigues Cortes**[1]*, **Ismael Henrique Silveira**[2], **Beatriz Fátima Alves de Oliveira**[3], **Michelle L. Bell**[4], **Washington Leite Junger**[1]

1 Institute of Social Medicine, State University of Rio de Janeiro, Rio de Janeiro, Brazil, 2 Institute of Collective Health, Federal University of Bahia, Salvador, Brazil, 3 Fiocruz Regional Office in Piauí, National School of Public Health, Oswaldo Cruz Foundation, Teresina, Brazil, 4 School of the Environment, Yale University, New Haven, CT, United States of America

* taisacortes@gmail.com

## Abstract

### Background

Several epidemiological studies have reported associations between ambient air pollution and mortality. However, relatively few studies have investigated this relationship in Brazil using individual-level data.

### Objectives

To estimate the short-term association between exposure to particulate matter <10 μm ($PM_{10}$) and ozone ($O_3$), and cardiovascular and respiratory mortality in Rio de Janeiro, Brazil, between 2012 and 2017.

### Methods

We used a time-stratified case-crossover study design with individual-level mortality data. Our sample included 76,798 deaths from cardiovascular diseases and 36,071 deaths from respiratory diseases. Individual exposure to air pollutants was estimated by the inverse distance weighting method. We used data from seven monitoring stations for PM10 (24-hour mean), eight stations for O3 (8-hour max), 13 stations for air temperature (24-hour mean), and 12 humidity stations (24-hour mean). We estimated the mortality effects of $PM_{10}$ and $O_3$ over a 3-day lag using conditional logistic regression models combined with distributed lag non-linear models. The models were adjusted for daily mean temperature and daily mean absolute humidity. Effect estimates were presented as odds ratios (OR) with their 95% confidence interval (CI) associated with a 10 μg/m3 increase in each pollutant exposure.

### Results

No consistent associations were observed for both pollutant and mortality outcome. The cumulative OR of $PM_{10}$ exposure was 1.01 (95% CI 0.99–1.02) for respiratory mortality and 1.00 (95% CI 0.99–1.01) for cardiovascular mortality. For $O_3$ exposure, we also found no

**Data Availability Statement:** The mortality data used in this study are protected by the General Personal Data Protection Law (No. 13.709) and the

National Health Council Resolution (No. 466/12) and cannot be made available due to potential identifying information (such as the residential address). Access to the mortality data can only be obtained after submitting a complete research project to the Research Ethics Committees at https://plataformabrasil.saude.gov.br/. After approval by the Research Ethics Committees, the mortality data can be requested from the Municipal Health Department of Rio de Janeiro (https://www.rio.rj.gov.br/web/sms/vigilancia-em-saude). Data regarding air pollution can be obtained from the DATA.RIO platform (https://www.data.rio/maps/PCRJ::qualidade-do-ar-dados-hor%C3%A1rios/about). Meteorological data can be obtained from the INMET meteorological database system at https://bdmep.inmet.gov.br/ and the PROTIM/CPTEC/INPE system at http://bancodedados.cptec.inpe.br/. All the files that are not protected under data protection laws, such as the environmental data and R codes, are available on GitHub (https://github.com/reprotc/Air_pollution_Rio).

**Funding:** This research was funded by the Coordination for the Improvement of Higher Education Personnel – CAPES (https://www.gov.br/capes/) (finance code 001), Foundation for Research Support of the State of Rio de Janeiro - FAPERJ (https://www.faperj.br/) (grant number grant numbers E-26/010.002131/2019 and E-26/200.966/2022), National Council of Technological and Scientific Development – CNPq (https://www.gov.br/cnpq/) (grant numbers 307495/2018-3, 406292/2018-3 and 315349/2021-2) and the Wellcome Trust (https://wellcome.org/) (grant number 216087/Z/19/Z). The funders had no role in study design, data collection and analysis, decision to publish, or preparation of the manuscript.

**Competing interests:** The authors have declared that no competing interests exist.

evidence of increased mortality for cardiovascular (OR 1.01, 95% CI 1.00–1.01) or respiratory diseases (OR 0.99, 95% CI 0.98–1.00). Our findings were similar across age and gender subgroups, and different model specifications.

## Conclusion

We found no consistent associations between the $PM_{10}$ and $O_3$ concentrations observed in our study and cardio-respiratory mortality. Future studies need to explore more refined exposure assessment methods to improve health risk estimates and the planning and evaluation of public health and environmental policies.

## Introduction

Air pollution is estimated to have contributed to approximately 7 million deaths in 2019 [1]. Mortality from respiratory and cardiovascular diseases are among the most commonly investigated health outcomes associated with air pollution [2]. Cardiovascular diseases are the leading cause of non-communicable diseases (NCDs) and accounted for nearly 18 million deaths in 2019 [3]. Chronic respiratory diseases also carry a significant burden of premature mortality and disability, and were the third leading cause of death in 2017 [4]. Non-communicable diseases are an important public health challenge worldwide, and air pollution is now ranked among their major risk factors [5,6].

Ambient air pollution consists of a complex and heterogeneous mixture of hazardous components. The health damage caused by air pollutants depends upon their levels and composition in the environment; the duration, frequency, and timing that individuals are exposed; and individual susceptibility [7]. Pollutant levels and composition are influenced not only by emission intensity and source (such as motor vehicles, industry, and biomass burning) but also by landscape features and meteorological processes [8,9]. Differential population vulnerabilities to air pollution might also be due to differences in socioeconomic conditions and demographic characteristics [10,11].

Since many of the factors affecting individual exposure and vulnerabilities can vary significantly between different populations and geographic areas, investigations into the health effects of air pollution are particularly relevant at regional and local level. In this study, we focus on two major pollutants, ambient particulate matter with an aerodynamic diameter $\leq 10~\mu m$ ($PM_{10}$) and ground-level ozone ($O_3$). Although several epidemiological studies have reported associations between ambient air pollution and mortality [2,12–14], relatively few studies in Brazil have investigated this relationship using individual-level data. In this study, we used a case-crossover design with individual-level exposure data to estimate the short-term association between exposure to $PM_{10}$ and $O_3$, and cardiovascular and respiratory disease mortality in Rio de Janeiro, Brazil. We also investigate effect modification by age and gender.

## Methods

### Mortality data

We carried out a time-stratified case-crossover study to investigate the association between short-term exposure to $PM_{10}$ and $O_3$, and cardiovascular and respiratory disease mortality in Rio de Janeiro (Brazil). Mortality data were obtained from the Municipal Health Department of Rio de Janeiro. We selected all deaths from cardiovascular (International Classification of Diseases, 10th Revision–ICD-10, codes I00-I99) and respiratory diseases (ICD-10, codes

J00-J99) that occurred in residents of the municipality of Rio de Janeiro between 2012 and 2017. Residential addresses were geocoded and validated using the method described in a previous study [15]. In brief, we used different automated methods for address matching and validation, and the performance of the geocoding process was assessed by manually reviewing a sample of addresses. Out of the total deaths (n = 132,863), approximately 86% (n = 113,876) were geocoded to street level, with a sensitivity and specificity of 87% and 98%, respectively.

Ethical approval for this study was obtained from the Research Ethics Committees of the Municipal Health Department of Rio de Janeiro and the State University of Rio de Janeiro. The mortality data were anonymized, and the authorization to carry out the research without the informed consent of subjects was given by the ethics committees.

### Environmental data

Hourly air pollution and meteorological data were obtained from the monitoring stations maintained by Rio de Janeiro's Municipal Secretariat for the Environment. Additional meteorological data were obtained from the Brazilian Institute of Meteorology and from airport weather stations maintained by Brazil's National Institute for Space Research.

We calculated daily concentrations of $PM_{10}$ ($\mu g/m^3$), maximum daily 8-hour $O_3$ ($\mu g/m^3$), mean daily temperature (C°), and mean relative humidity (%). For the computation of pollutant daily averages, the availability of at least 75% of the hourly values (for each monitor) was required. We only included monitoring station data in cases where less than 15% of the study period data was missing (7 stations for $PM_{10}$, 8 for $O_3$, 13 for temperature, and 12 for relative humidity). Absolute humidity was estimated using relative humidity and temperature data, as previously described [16]. The location of the meteorological and air quality monitoring stations used in the study are shown in the S1 Fig.

The imputation of missing data was performed using a method based on the expectation-maximization algorithm implemented in the mtsdi package [17]. Missing values were estimated using the covariance matrix of the stations' data. The series temporal component (trend and seasonality) was modeled using natural cubic splines with 5 degrees of freedom per year. The amount of missing data imputed in each time series is presented in S1 Table.

### Exposure assessment

Individual-level exposures to air pollutants and meteorological variables were estimated using the inverse distance weighting (IDW) method. In this method, the values at unmeasured locations are estimated by averaging the weighted sum of values from the nearest neighbors (monitoring stations) within a search radius. The weights assigned to each neighboring value are expressed as a function of the inverse distance (between the measured and unmeasured locations) raised to a non-negative power [18].

Leave-one-out cross-validation was carried out to determine the optimal power value, the number of nearest neighbors, and the search radius. For each day, we selected the set of parameters that minimized the prediction root mean square error (RMSE) [19]. The power of distance was evaluated using five values from 1 to 3, and the number of neighbors ranged from 1 to the maximum number of stations (7, 8, 12, or 13). Restricting the search radius to 10km, 15km, or 20km did not improve interpolation results. Although we observed better results for air pollutants using the 5km radius, these data were only used in the sensitivity analysis due to the high number of grids with missing data. The root mean square error (RMSE) and mean absolute error (MAE) values of cross-validation for the different sets of parameters are summarized in S2 Table.

Daily weighted averages of $PM_{10}$, $O_3$, and meteorological variables were interpolated into a one square kilometer grid [20]. Individual exposure estimates were calculated through the average of the interpolated values within a 1-kilometer buffer of each residential address.

## Statistical analysis

The short-term mortality effects of $PM_{10}$ and $O_3$ were estimated using conditional logistic regression models combined with distributed lag non-linear models [21]. In line with most previous studies [12], we assumed a linear relationship between air pollution concentration and mortality outcomes. We considered the effects of each pollutant exposure on the day of death (lag 0) and exposures from 1 to 3 days preceding death (lag 1 to lag 3). The lag–response dimension was defined using a natural cubic spline function with three degrees of freedom.

We used a time-stratified case-crossover study with control days selected on the same day of the week and in the same month and year of death [22]. Time-invariant confounders were adjusted by design, as a consequence of self-matching. Time-variant factors, such as daily temperature and daily humidity, were considered potential confounders (as depicted in the causal directed acyclic graph (DAG) in S1 Appendix). We used a cross-basis function to account for the potential nonlinear confounding effects of temperature [21]. The exposure-response dimension of the cross-basis function for temperature was defined using a natural cubic spline with two internal knots (at the 50th and 75th percentiles), while the lag dimension was modeled with a natural cubic spline with two degrees of freedom and a maximum lag of up to 3 days. Daily absolute humidity (lag 0–3 average) was adjusted using a natural cubic spline with one internal knot. We controlled for absolute humidity rather than relative humidity, since the former is a direct measure of water vapor in the air, regardless of air temperature, thus potentially minimizing over-fitting and multicollinearity [23,24]. The number of knots and their respective locations were selected based on the Akaike Information Criteria [25]. More details about our modeling choices are provided in S1 Appendix.

Estimates of lag-specific (total) and cumulative effects are presented as odds ratios (OR), and their 95% confidence interval (CI), associated with a 10 μg/m3 increase in each pollutant exposure. We performed stratified analysis by gender (female, male) and age group. In the age-stratified analysis, the deaths were categorized into two age groups ($<64$ years and $\geq 65$ years).

We carried out several sensitivity analyses to examine the robustness of our model estimates. We investigated lag periods of 5 and 10 days for $PM_{10}$ and $O_3$, and 7 and 21 days for temperature. We also checked that the results were robust to different adjustments for humidity (a model with relative humidity and one without any humidity adjustment) and for temperature. In addition, since pollutant concentrations may have high intra-urban spatial variation, we restricted our analysis to deaths whose geocoded address was located within 10km and 5km of a monitoring station. Finally, we estimated the cumulative effect over a 3-day lag using a time series design and Poisson regression models. The models were adjusted for daily temperature, absolute humidity, long-term trends (natural cubic spline of time with 5 degrees of freedom per year), and day of the week (as an indicator variable).

All analyses were conducted using R 4.1.0. We used different R packages, including the packages survival [26] and dlnm [21] to fit the main models, and the package gstat for interpolation [27]. All the files that are not protected under data protection laws, such as the environmental data and R codes, are available at https://github.com/reprotc/Air_pollution_Rio.

## Results

During the study period (2012–2017), 90,897 deaths from cardiovascular disease and 41,966 deaths from respiratory diseases were recorded. We excluded those deaths with non-geocoded

**Table 1. Summary statistics of pollutant concentrations and meteorological variables in Rio de Janeiro, Brazil (2012–2017).**

|  | Mean | SD | Median | IQR | Minimum | Maximum |
|---|---|---|---|---|---|---|
| PM$_{10}$ (μg/m$^3$) | 37.76 | 13.57 | 35.48 | 16.95 | 11.46 | 107.76 |
| O$_3$ (μg/m$^3$) | 54.31 | 19.64 | 51.97 | 27.01 | 11.32 | 143.97 |
| Daily temperature (°C) | 25.36 | 3.16 | 25.15 | 4.73 | 15.51 | 33.97 |
| Daily relative humidity (%) | 71.84 | 8.99 | 72.23 | 11.90 | 44.56 | 94.72 |

PM$_{10}$: Particulate matter ≤10 μm; O$_3$: Ozone; SD: Standard deviation; IQR: Interquartile range.

addresses and lagged exposure prior to 2012. A total of 76,798 deaths (84%) due to cardiovascular disease and 36,071 deaths (86%) due to respiratory disease were thus included in our analysis. S3 Table presents descriptive statistics for the study sample and the excluded participants.

Summary statistics for air pollutants and meteorological variables are provided in Table 1. The means (and standard deviations) of air pollutants were 37.8 (13.6) μg/m$^3$ for PM$_{10}$ and 54.3 (19.7) μg/m$^3$ for O$_3$.

The lag-specific mortality association for a 10 μg/m$^3$ increase in air pollution exposure is shown in Fig 1 (and S4 Table). For PM$_{10}$, the results indicate a slight increase in respiratory mortality at lag 0, lag 2 and lag 3. The PM$_{10}$ effect estimate for cardiovascular mortality was slightly higher at lag 1 and lag 2. As with ozone, we found inverse associations with respiratory mortality over the 3-day lag, while for cardiovascular mortality most point estimates were close to the null value.

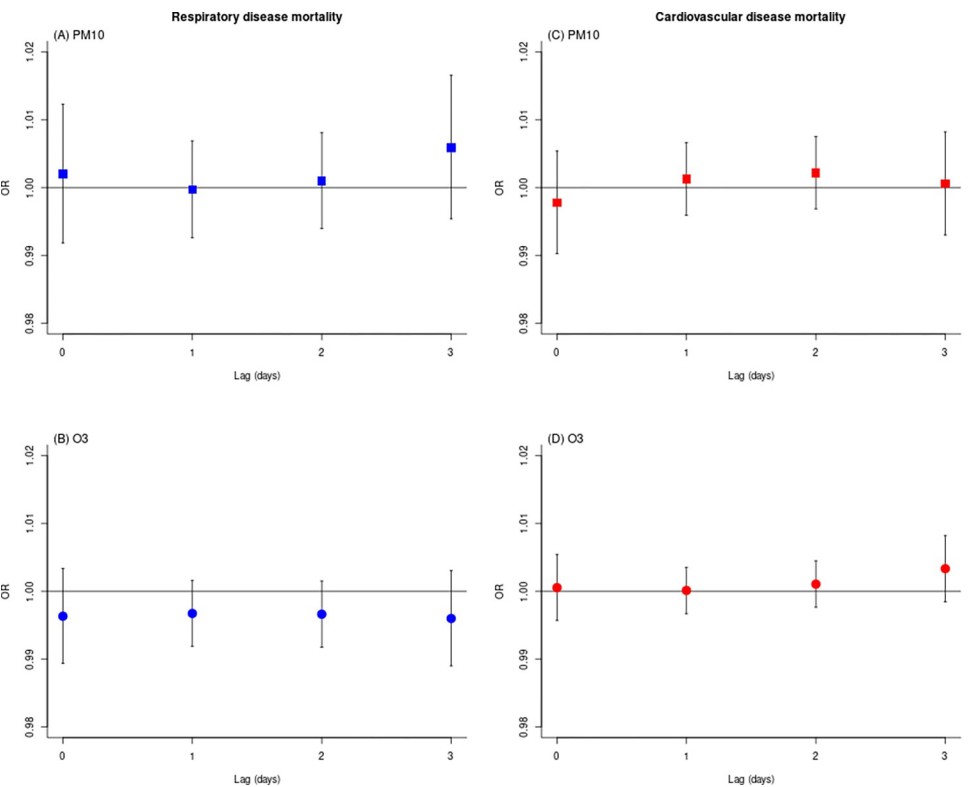

**Fig 1.** Lag-specific association of short-term exposure to PM10 and O3, and respiratory (A-B) and cardiovascular disease mortality (C-D), Rio de Janeiro (Brazil) (2012–2017). PM10: Particulate matter ≤10 μm; O3: Ozone; OR: Odds ratio.

**Table 2. Cumulative associations of short-term exposure to PM$_{10}$ and O$_3$, and cardiovascular and respiratory mortality, Rio de Janeiro (Brazil), (2012–2017).**

| Pollutant | Subgroup | Respiratory disease mortality OR (95% CI) | Cardiovascular disease mortality OR (95% CI) |
|---|---|---|---|
| PM$_{10}$ | All | 1.01 (0.99–1.02) | 1.00 (0.99–1.01) |
| | Age ≥ 65 years | 1.01 (0.99–1.02) | 1.00 (0.99–1.01) |
| | Age 0–64 years | 1.01 (0.98–1.05) | 1.00 (0.98–1.02) |
| | Female | 1.01 (0.99–1.03) | 1.01 (0.99–1.02) |
| | Male | 1.01 (0.99–1.03) | 1.00 (0.98–1.01) |
| O$_3$ | All | 0.99 (0.98–1.00) | 1.01 (1.00–1.01) |
| | Age ≥ 65 years | 0.99 (0.98–1.00) | 1.00 (1.00–1.01) |
| | Age 0–64 years | 0.97 (0.95–0.99) | 1.01 (0.99–1.02) |
| | Female | 0.99 (0.97–1.00) | 1.01 (1.00–1.02) |
| | Male | 0.98 (0.97–1.00) | 1.00 (0.99–1.01) |

OR: Odds ratio for a 10 μg/m3 increase in each pollutant exposure.; 95% CI: 95% confidence interval; PM$_{10}$: Particulate matter ≤10 μm; O$_3$: Ozone. Models were performed using case-crossover design and adjusted for daily mean temperature and absolute humidity.

No consistent cumulative associations were observed for both pollutants and mortality outcomes. As shown in Table 2, the overall estimate of the cumulative effect over a 3-day lag for a 10 μg/m$^3$ increase in PM$_{10}$ exposure was 1.01 (95% CI 0.99–1.02) for respiratory mortality and 1.00 (95% CI 0.99–1.01) for cardiovascular mortality. Neither did we find evidence of increased mortality for cardiovascular (OR 1.01, 95% CI 1.00–1.01) or respiratory diseases (OR 0.99, 95% CI 0.98–1.00) for O$_3$.

Table 2 shows the air pollution-mortality associations according to the age and gender subgroups. Although we found no evidence of effect modification, the association between PM$_{10}$ and mortality outcomes was positive (although not significant) for most subgroups.

Sensitivity analyses using different model specifications are presented in S5 Table. We found similar results when we removed subjects whose homes were located 10km or 5km away from the nearest monitoring station. Findings also did not change when using different lag periods for pollutant exposures, and different adjustments for temperature and humidity. However, the analyses using the time series design and Poisson regression models show an increase in respiratory mortality associated with changes in PM$_{10}$ concentration. In addition, we found positive associations between both pollutants (PM$_{10}$ and O$_3$) and cardiovascular disease (CVD) mortality (S5 Table).

## Discussion

In this study, we investigated the effects of ambient PM$_{10}$ and O$_3$ exposure on mortality from respiratory and cardiovascular diseases among residents of Rio de Janeiro, Brazil. We found no consistent association between the pollutant concentrations observed in our study and cardio-respiratory mortality. Moreover, we found no evidence of increased mortality effects from air pollution among the age and gender subgroups.

Our results differ from most of the literature regarding the mortality effects of air pollution [12,13,28,29]. In a systematic review and meta-analysis of 196 studies assessing the short-term mortality effects of air pollution on various continents, Orellano et al. [12] found significant positive associations between ambient levels of PM$_{10}$ and cardiovascular and respiratory mortality, while for O$_3$ the authors only found an association with all-cause mortality. A multi-city study in Latin America also found positive associations between ambient PM$_{10}$ levels and cardio-respiratory mortality in most of the cities, including Rio de Janeiro, Porto Alegre, and São

Paulo in Brazil [29]. In Rio de Janeiro, $PM_{10}$ concentrations were associated with a 2.14% (95% CI 1.41–2.87) increase in respiratory mortality and 0.51% (95% CI 0.06–0.96) increase in cardiovascular mortality, while $O_3$ was only associated with cardiovascular mortality [29]. In contrast, a case-crossover study carried out in São Paulo found that higher concentrations of $O_3$ was only associated with respiratory mortality [30].

Although some studies have produced similar results to our own, especially for $O_3$ [12,29,30], evidence from several epidemiological and toxicological studies (both in humans and animals) supports the relationship between air pollution exposure and cardio-respiratory morbidity and mortality [31]. While each pollutant has its own specific toxic properties, increasing evidence indicates oxidative stress and inflammation as key processes underlying the health effects of air pollution [6,32,33]. Air pollutants such as particulate components (PM) and ozone can cause oxidative stress within the lungs, leading to an inflammatory cascade in tissues which can itself lead to systemic inflammation and cell damage [33]. These responses can impact the lung and other organs, either directly or as a secondary effect [34]. Various acute effects of air pollution on the respiratory system have been described in the literature, including the onset and worsening of asthma, and chronic obstructive pulmonary disease (COPD) admissions and mortality. Other recent findings indicate that short-term exposure to major air pollutants (including $PM_{2.5-10}$ and ozone) can increase the risk of hospitalization and mortality due to ischemic heart diseases [6], heart failure and ischemic stroke [35], particularly in high-risk populations.

Several factors could account for the heterogeneity of the air pollution-mortality association across studies, including differences in research methods, chemical composition of pollutants, and population characteristics [7,10–12,28]. Nonetheless, we believe that the primary reason for the different findings in our study, compared to others, arises from the characteristics of our air pollution data. The daily variability in pollutant concentrations in this study was very low, such that the concentration difference between event and control days had a mean value close to zero (S6 Table). As discussed by Kunzli and Schindler [36], similar exposure levels between case and control periods can provide low statistical power for the detection of pollutant effects in case-crossover studies. Our sensitivity analysis, using a Poisson time series, supports this hypothesis, since we found consistent positive associations between $PM_{10}$ and respiratory disease mortality, and between both pollutants ($PM_{10}$ and $O_3$) and CVD mortality (S5 Table). However, compared to case-crossover studies, time series studies are more prone to residual confounding due to seasonality and long-term trends compared to case-crossover studies, and thus potential bias might still exist.

Another study limitation is the use of fixed monitoring stations to estimate air pollution exposure. We attempted to reduce exposure error by capturing more of the spatial variability in air pollutants through an interpolation method (IDW). However, this still represents a coarse exposure approach, since it does not account for spatial heterogeneity due to land use (i.e., roads) and meteorological conditions (i.e., wind, solar radiation). Our methods also did not consider the essential determinants of individual exposure, such as daily mobility and indoor air pollution. This may have led to independent and non-differential measurement errors [37], since the influence of individual activity patterns and time spent indoors/outdoors may have been partially "adjusted" due to self-matching in the case-crossover design (S1 Appendix).

However, the performance of spatial interpolation methods such as IDW may be limited in settings with a sparse monitoring network [38]. In our study, the average distance between the nearest monitoring stations and residential addresses was approximately 6 kilometers for $PM_{10}$, 5 kilometers for $O_3$ and 4 kilometers for the meteorological variables. Although we found similar results when excluding deaths whose geocoded address was located within 5km

away from the nearest monitoring station (S5 Table), bias due to exposure measurement error is likely to have contributed to our findings.

While the mortality data in Brazil has been assessed as good quality [39], the exclusion of deaths due to missing geocoded addresses (14%) could also have affected our results. Our excluded sample had a higher proportion of people of color (54% vs. 37%) and lower educational attainment (74% vs. 61%), which are usually markers for residing in lower income neighborhoods, those with greater segregation and where the cartographic base may be of lower quality. Although we believe that the selection bias magnitude was not high, its impact depends on whether these locations also have higher levels of air pollution or more significant exposure variability.

In conclusion, we found no consistent associations between the $PM_{10}$ and $O_3$ concentrations observed in our study and cardio-respiratory mortality. To the best of our knowledge, this is the first study using individual-level exposure estimates that has investigated the association between $PM_{10}$ and $O_3$ and cardio-respiratory mortality among residents of all ages in Rio de Janeiro, Brazil.

The lack of high spatial and temporal resolution air pollution data remains a significant challenge for air pollution research. Future investigations using other study designs should benefit from more refined exposure assessments [40], including satellite data, air pollution modeling, and individual mobility patterns. Enhancing exposure assessment is essential for improving health risk estimates. Adequate monitoring of population exposure to air pollutants is also crucial for planning and evaluating public health and sustainable development policies.

## Supporting information

**S1 Fig. Map of the city of Rio de Janeiro area depicting the locations of weather and air quality monitoring stations used in the study.** The hottest spots (in red) indicate a high density of deaths within a 500-meter buffer zone.
(TIF)

**S1 Table. Percentage of imputed missing data.** Stations with more than 25% missing values in a year and/or $> 15\%$ missing data over the study period were excluded.
(PDF)

**S2 Table. Root mean square error (RMSE) and mean absolute error (MAE) values of cross-validation for the different sets of inverse distance weighting parameters.**
(PDF)

**S3 Table. Descriptive statistics of the study sample and excluded participants.**
(PDF)

**S4 Table. Lag-specific association of short-term exposure to $PM_{10}$ and $O_3$, and cardiovascular and respiratory mortality, Rio de Janeiro (Brazil), (2012–2017).** OR: odds ratio; 95% CI: 95% confidence interval; Models were performed using case-crossover design and adjusted for temperature and relative humidity.
(PDF)

**S5 Table. Sensitivity analyses of the short-term mortality effects of $PM_{10}$ and $O_3$.** OR: Odds ratio for a 10 μg/m$^3$ increase in each pollutant exposure; 95% CI: 95% confidence interval. ** Risk Ratio and 95% CI estimated using a time series design and Poisson regression models adjusted for mean daily temperature, mean daily absolute humidity, seasonality, and day of the week.
(PDF)

**S6 Table. Pollutant concentration differences between event days and average concentrations over the control period in Rio de Janeiro, Brazil (2012–2017).** * Differences between daily concentrations of each pollutant on event days (day of death) and average concentrations over the control period. **Difference between pollutant concentrations in event and control days using all the mortality data (geocoded and non-geocoded) and daily average exposure data (all stations without imputation and IDW interpolation). $PM_{10}$: Particulate matter $\leq 10$ μm; $O_3$: Ozone; SD: Standard deviation; IQR: Interquartile range.
(PDF)

**S1 Appendix. Modeling choices.**
(PDF)

## Author Contributions

**Conceptualization:** Taísa Rodrigues Cortes, Ismael Henrique Silveira, Beatriz Fátima Alves de Oliveira, Michelle L. Bell, Washington Leite Junger.

**Data curation:** Taísa Rodrigues Cortes.

**Formal analysis:** Taísa Rodrigues Cortes.

**Funding acquisition:** Michelle L. Bell, Washington Leite Junger.

**Methodology:** Taísa Rodrigues Cortes, Ismael Henrique Silveira, Beatriz Fátima Alves de Oliveira, Michelle L. Bell, Washington Leite Junger.

**Validation:** Ismael Henrique Silveira.

**Visualization:** Taísa Rodrigues Cortes, Ismael Henrique Silveira.

**Writing – original draft:** Taísa Rodrigues Cortes.

**Writing – review & editing:** Taísa Rodrigues Cortes, Ismael Henrique Silveira, Beatriz Fátima Alves de Oliveira, Michelle L. Bell, Washington Leite Junger.

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
