## [Decision Letter · Decision Letter 0]

27 Sep 2022

PONE-D-22-17313Short-term association between ambient air pollution and cardio-respiratory mortality in Rio de Janeiro, Brazil.PLOS ONE

Dear Dr. Cortes,

Thank you for submitting your manuscript to PLOS ONE. After careful consideration, we feel that it has merit but does not fully meet PLOS ONE’s publication criteria as it currently stands. Therefore, we invite you to submit a revised version of the manuscript that addresses the points raised during the review process.

The comments by Reviewer 1 related to measurement error and the validity of your exposure estimates are a major concern to me as well, and should be addressed more fully in your discussion section, and potentially adding additional detail in Methods or Results. While you do discuss the issues of measurement error already, the authors should attempt to further alleviate reviewer and reader concerns about the fewer stations available for pollutants, distance between residential addresses and stations for the deaths included, etc. It may be helpful to provide a map of the location of the stations for each exposure type along side the count of deaths by appropriate sub-city Census unit (so that death data remains de-identifiable) and calculate the average distances between deaths and stations, for example. These are just examples and the authors may decide on a better way to address these concerns. However, I do not agree that these issues should result in a rejection, if the authors are able to address them in more detail. I feel that this paper despite the null findings adds to the evidence base on this topic and represents sound statistical methods/analysis. 

We look forward to receiving your revised manuscript.

Kind regards,

Kristina Weis Kintziger, PhD, MPH

Academic Editor

PLOS ONE

Journal Requirements:

This research was funded by the Coordination for the Improvement of Higher Education Personnel – CAPES (https://www.gov.br/capes/ ) (finance code 001), Foundation for Research Support of the State of Rio de Janeiro - FAPERJ (https://www.faperj.br/)(grant number E-26/202.756/2018), National Council of Technological and Scientific Development – CNPq (https://www.gov.br/cnpq/)(grant numbers 307495/2018-3 and 406292/2018-3) and the Wellcome Trust (https://wellcome.org/) (grant number 216087/Z/19/Z). 

Additional Editor Comments:

Additional things that should be addressed in the revision:

1. Please include the direct website link on Github for the available code and data.

2. In first paragraph of results, please provide the % of total deaths of each type that were included after all exclusions were made.

Reviewers' comments:

Reviewer's Responses to Questions

**Comments to the Author**

1. Is the manuscript technically sound, and do the data support the conclusions?

Reviewer #1: No

Reviewer #2: Yes

2. Has the statistical analysis been performed appropriately and rigorously? 

Reviewer #1: No

Reviewer #2: Yes

3. Have the authors made all data underlying the findings in their manuscript fully available?

Reviewer #1: Yes

Reviewer #2: Yes

4. Is the manuscript presented in an intelligible fashion and written in standard English?

Reviewer #1: Yes

Reviewer #2: Yes

5. Review Comments to the Author

Reviewer #1: The authors explored the associations between short-term exposures to PM10 and ozone and cardiovascular and respiratory mortality in Rio de Janeiro. They used a time-stratified case-crossover study design adjusting for temperature and humidity. The air pollution and meteorological data were obtained from the monitoring stations. The authors did not detect increased cardiovascular and respiratory mortality associated with both PM10 and ozone.

The big question for me is the high degree of exposure measurement error due to the use of monitored data in the context of case crossover design. Generally, the weather station will be established far away from the urbanized areas, and the meteorological data might not be consistent to the city. This introduces large exposure error, which may attenuate the resulting effect estimates or even results in the regression coefficients to have the wrong sign. The authors should assess the validity.

Reviewer #2: The comments are as attached in PDF form. Overall, it is a good attempt as I rarely seen researchers analyzed for individual level data, and usually people will only consider for pool health impacts.

6. PLOS authors have the option to publish the peer review history of their article (what does this mean?). If published, this will include your full peer review and any attached files.

Reviewer #1: No

Reviewer #2: No

---

## [Author Response · Author response to Decision Letter 0]

11 Nov 2022

Authors' responses: 

We thank the editor and both reviewers for their time and helpful suggestions for our paper. We have provided answers to each of your questions below.

Editor: 

The comments by Reviewer 1 related to measurement error and the validity of your exposure estimates are a major concern to me as well, and should be addressed more fully in your discussion section, and potentially adding additional detail in Methods or Results. While you do discuss the issues of measurement error already, the authors should attempt to further alleviate reviewer and reader concerns about the fewer stations available for pollutants, distance between residential addresses and stations for the deaths included, etc. It may be helpful to provide a map of the location of the stations for each exposure type along side the count of deaths by appropriate sub-city Census unit (so that death data remains de-identifiable) and calculate the average distances between deaths and stations, for example. These are just examples and the authors may decide on a better way to address these concerns. However, I do not agree that these issues should result in a rejection, if the authors are able to address them in more detail. I feel that this paper despite the null findings adds to the evidence base on this topic and represents sound statistical methods/analysis. 

Authors' response: 

We agree that the limited availability of exposure data is an important limitation of our study. According to your suggestions, we added information on distance between residential addresses and monitoring stations, as well as a map displaying the locations of the pollutant stations and deaths density (Figure S1). The new version of our manuscript also includes the additional comment regarding measurement error: 

“However, the performance of spatial interpolation methods such as IDW may be limited in settings with a sparse monitoring network [38]. In our study, the average distance between the nearest monitoring stations and residential addresses was approximately 6 kilometers for PM10, 5 kilometers for O3 and 4 kilometers for the meteorological variables. Although we found similar results when excluding deaths whose geocoded address was located within 5km away from the nearest monitoring station (S5 Table), bias due to exposure measurement error is likely to have contributed to our findings” (p.10, lines 274-280).

We hope that these revisions will be satisfactory.

Additional Editor comments: 

Please include the direct website link on Github for the available code and data.

Authors' response: 

The github (https://github.com/reprotc/Air_pollution_Rio) link has been added to the manuscript.

In first paragraph of results, please provide the % of total deaths of each type that were included after all exclusions were made. 

Authors' response: 

We changed the paragraph in accordance with your recommendation (p. 6, lines 177-178).

Responses to reviewers:

Reviewer #1: The authors explored the associations between short-term exposures to PM10 and ozone and cardiovascular and respiratory mortality in Rio de Janeiro. They used a time-stratified case-crossover study design adjusting for temperature and humidity. The air pollution and meteorological data were obtained from the monitoring stations. The authors did not detect increased cardiovascular and respiratory mortality associated with both PM10 and ozone. The big question for me is the high degree of exposure measurement error due to the use of monitored data in the context of case crossover design. Generally, the weather station will be established far away from the urbanized areas, and the meteorological data might not be consistent to the city. This introduces large exposure error, which may attenuate the resulting effect estimates or even results in the regression coefficients to have the wrong sign. The authors should assess the validity.

Authors' response: 

As previously mentioned, we provide additional information on the location of monitoring sites and the distribution of deaths in our study area, as well as a new paragraph discussing measurement error (please, see S1 Figure and p.10 lines 274-280). 

Unfortunately, it was not possible to implement statistical approaches to quantify the impacts of measurement error on our point estimates and confidence intervals. However, these untouched issues relate to exposure assessment will be the subject of our future investigation. 

Reviewer #2: It is better if you could use the latest reference (line 55)

Authors' response: We appreciate the reviewer's suggestion. In the new version of the manuscript, we added the most recent report on the health impacts of air pollutants.

Reviewer #2: Explain a little bit about how you get the sensitivity and specificity percentage 

Authors' response: We have added the following clarifications regarding the sensitivity and specificity of geocoding: “Residential addresses were geocoded and validated using the method described in a previous study [15]. In brief, we used different automated methods for address matching and validation, and the performance of the geocoding process was assessed by manually reviewing a sample of addresses. Out of the total deaths (n=132,863), approximately 86% (n=113,876) were geocoded to street level, with a sensitivity and specificity of 87% and 98%, respectively ” (p. 3, lines 89-93). 

Reviewer #2: Why temperature and relative humidity variables were taken, and not described in the title, abstract and introduction (lines 100-102)

Authors' response: In response to the reviewer's question, we added to the abstract information about the meteorological variables used in our study. Since these variables were thought to be confounders, we assumed it would be appropriate to report them only in the abstract and methods section, as suggested by the STROBE.

Reviewer #2: Is this sufficient to be extrapolated to the whole Rio de Janeiro? (lines 103-104)

Authors' response: We agree that the representativeness of monitoring data is a important issue. In the new version of the manuscript, we attempted to address your question by including additional information on the stations density and futher discussion on exposure measurement error (please, see p.10 lines 274-280).

We hope that these revisions will be satisfactory.

Reviewer #2: Line 136-137: Why you consider only until lag 3? I found many publications with significant findings at Lag 6. 

Authors' response: To select the lag length, we first examined the relationship between air pollution and mortality using a lag of 15 days, as described in the supplemental material (Appendix S1). While the main results were presented with a maximum lag of three days, we found very similar results when using different lag periods (such as lag 0-5, lag 0-10 days), as described in our sensitivity analyses.

Reviewer #2: why considered as non significant? it is a protective factor

Authors' response: We revised the text based on your comment (Please, see p. 7, line 190)

Reviewer #2: Line 230-231: Are they using Lag1-3?

Authors' response: We agree that methodological aspects such as the lag duration should be considered when comparing risk estimates between studies. We believe, however, that our choices for the exposures lags does not appear to be the an explanation for our findings, as we found similar results using different lag structures (please, see S5 Table).

Reviewer #2: Table 1: μg/m3

Authors' response: We appreciate the reviewer correction and have modified the text accordingly.

---

## [Editor Report · Decision Letter 1]

13 Dec 2022

PONE-D-22-17313R1Short-term association between ambient air pollution and cardio-respiratory mortality in Rio de Janeiro, Brazil.PLOS ONE

Dear Dr. Cortes,

Thank you for submitting your manuscript to PLOS ONE. After careful consideration, we feel that it has merit but does not fully meet PLOS ONE’s publication criteria as it currently stands. Therefore, we invite you to submit a revised version of the manuscript that addresses the points raised during the review process.

I am unable to make changes to authorship. Please add the new author as requested on 12/1/2022, in the location of your choosing and provide appropriate contact information, as needed. Then, resubmit the final version. Thank you for addressing all of my and the reviewers' concerns.

We look forward to receiving your revised manuscript.

Kind regards,

Kristina Weis Kintziger, PhD, MPH

Academic Editor

PLOS ONE

Journal Requirements:

Additional Editor Comments (if provided):

Thank you for the revisions to your manuscript. All requests have been adequately addressed. Please add the additional co-author with appropriate contact or other information as requested on 12/1 by the corresponding author and resubmit a final version.
---

## [Author Response · Author response to Decision Letter 1]

5 Jan 2023

Editor: 

I am unable to make changes to authorship. Please add the new author as requested on 12/1/2022, in the location of your choosing and provide appropriate contact information, as needed. Then, resubmit the final version. Thank you for addressing all of my and the reviewers' concerns.

Additional Editor comments: 

Thank you for the revisions to your manuscript. All requests have been adequately addressed. Please add the additional co-author with appropriate contact or other information as requested on 12/1 by the corresponding author and resubmit a final version.

Authors' response: We thank you for your time and suggestions and for providing us with the opportunity to submit a final version of our manuscript. We added the new author with appropriate contact information. In addition, we have made few minor modifications to the manuscript, primarily the correction of typographical errors (lines 114, 205, 300) and the addition of citations for a few statements (line 264).

Journal Requirements:

Authors' response: We checked the reference list and found no retractions. However, we corrected the manuscript title of reference 34, which was incomplete.

---

## [Editor Report · Decision Letter 2]

25 Jan 2023

Short-term association between ambient air pollution and cardio-respiratory mortality in Rio de Janeiro, Brazil.

PONE-D-22-17313R2

Dear Dr. Cortes,

We’re pleased to inform you that your manuscript has been judged scientifically suitable for publication and will be formally accepted for publication once it meets all outstanding technical requirements.

Kind regards,

Kristina Weis Kintziger, PhD, MPH

Academic Editor

PLOS ONE

Additional Editor Comments (optional):

Thank you for addressing all of the reviewers', editor's, and journal's requests. I have submitted a decision of "Accept".
---

## [Editor Report · Acceptance letter]

7 Feb 2023

PONE-D-22-17313R2 

Short-term association between ambient air pollution and cardio-respiratory mortality in Rio de Janeiro, Brazil. 

Dear Dr. Cortes:

I'm pleased to inform you that your manuscript has been deemed suitable for publication in PLOS ONE. Congratulations! Your manuscript is now with our production department. 

Kind regards, 

on behalf of

Dr. Kristina Weis Kintziger 

Academic Editor

PLOS ONE